environmental science/green chemistry/ power and energy systems

alumina extraction, coal gangue, hydro-chemical process, phase transformation

**Author for correspondence:**
Changsheng Shi
e-mail: northinstitute@yeah.net

This article has been edited by the Royal Society of Chemistry, including the commissioning, peer review process and editorial aspects up to the point of acceptance.

# Extraction of alumina from alumina rich coal gangue by a hydro-chemical process

Quancheng Yang[1,2], Fan Zhang[1], Xingjian Deng[1], Hongchen Guo[1], Chao Zhang[1], Changsheng Shi[1] and Ming Zeng[2]

[1]Department of Environmental Engineering, North China Institute of Science and Technology, Beijing 101601, People's Republic of China
[2]School of Chemical and Environmental Engineering, China University of Mining and Technology (Beijing), Beijing 100083, People's Republic of China

(iD) QY, 0000-0003-0808-0829

Vast quantities of gangue from coal mining and processing have accumulated over the years and caused significant economic and environmental problems in China. For high added-value utilization of alumina rich coal gangue (ARCG), a mild hydro-chemical process was investigated to extract alumina. The influences of NaOH concentration, mass ratio of alkali to gangue, reaction temperature and reaction time were systematically studied. An alumina extraction rate of 94.68% was achieved at the condition of NaOH concentration 47.5%, alkali to gangue ratio of 6, reaction temperature of 260°C and reaction time of 120 min. The obtained leaching residues were characterized through X-ray diffraction, scanning electron microscopy and energy-dispersive spectrometer. Research confirmed that kaolinite the main alumina-bearing phase of ARCG can be decomposed and transformed to $Na_8Al_6Si_6O_{24}(OH)_2(H_2O)_2$ and $Ca_2Al_2SiO_6(OH)_2$ at relatively low temperature and short reaction time. Additionally, $Na_8Al_6Si_6O_{24}(OH)_2(H_2O)_2$ and $Ca_2Al_2SiO_6(OH)_2$ are unstable and will transform to alumina-free phase $NaCaHSiO_4$ under the optimal conditions, which is the major reason for high alumina extraction rates.

# 1. Introduction

Coal gangue is an industrial solid waste produced by coal mining and processing [1–3]. The average production of gangue accounts for 10–15% of raw coal production [4,5], making coal gangue one of the biggest solid wastes in China. At present, utilization ratio of coal gangue can reach about 60% in some countries. There are still huge amounts of coal gangue being dumped in coal reject hills

and posing severe environmental and social problems [6–8]. For the present, large-scale utilization technologies of coal gangue are mainly characterized by low added-value such as making cement and bricks, power generation and paving roads [9,10]. To realize high added-value utilization of coal gangue, numerous methods have been extensively studied [11–17].

In recent years, a special coalfield has been found in the Inner Mongolia and Northern Shanxi of China, which is characterized by high content of alumina. The coal gangue generated from this area is thereby rich in alumina ($Al_2O_3$, 40–50 wt%), which is almost the same as some middle and low-grade bauxite [18]. Alumina mainly extracted from bauxite is widely used in catalysts, adsorbents, ceramics, refractory, metallurgy and many other areas [19]. As the largest producer and consumer of aluminium in the world, China has been in great demand for alumina. However, due to the lack of bauxite resources, the import dependency of bauxite had exceeded 50% in China [20]. Therefore, the extraction of alumina from non-bauxite resources has become a consensus in China. Alumina rich coal gangue (ARCG) can be seen as a potential alternative alumina resource. Nevertheless, conventional alumina extraction method such as Bayer process is not suitable for treating ores with low alumina silica ratio, because the silica and alumina would be simultaneously dissolved and form sodium alumino-silicate hydrate precipitation [21]. To solve this problem, a large number of studies for the extraction of alumina from low-grade aluminium resources have been reported. The methods are mainly based on the hydrometallurgical processes using acid or alkali as the reaction medium [22,23].

For acid leaching, a pre-activated process including mechanical activation and thermal activation is always needed to make alumina-containing phase decomposition, otherwise only amorphous alumina can be leached and the extraction ratio would be limited [2–4,15]. In acid process, using sulfuric acid, hydrochloric acid or nitric acid as solvent, the impurity elements such as iron oxide, calcium oxide and titanium oxide will be simultaneously leached in leaching liquor, which makes the impurity removal process more complicated [19,21]. In addition, acids are normally volatilizable and expensive, resulting in relatively uneconomic extraction process. Moreover, acid leaching method may cause corrosion on the metal pipes and equipment, which limits its practical applications [24,25].

Alkali process primarily contains sintering and hydro-chemical process. In the sintering process, alumina-containing materials are mixed with limestone and sintered, transforming silica into dicalcium silicate and alumina into calcium aluminate. Then, alumina can be separated from silica efficiently in followed sodium carbonate leaching process [26]. However, the sintering method is not suitable for large-scale industrialization due to some problems hard to solve, such as high energy consumption, large-waste residues production and narrow range of sintering reaction temperature [27].

The hydro-chemical process was put forward by Soviet scientists for the first time and can fix silica by forming $NaCaHSiO_4$ which contains no alumina. The hydro-chemical process has been employed to extract alumina from low-grade alumina-bearing materials, containing red mud, nepheline and fly ash [28–30]. Our previous work showed that an ideal alumina extraction rate from fly ash (more than 90%) could be achieved by a mild hydro-chemical process [31]. Based on the research, our team built a demonstration project with annual production of 10 000 tonnes alumina in 2014, which achieved the expected results [32]. The results confirmed that hydro-chemical process can obviously enhance the mass transfer process and promote mineral decomposition, comparing with the sintering process.

Considering Al–O octahedron and Si–O tetrahedron structure similarities between coal gangue and mullite (the main alumina-bearing phase in fly ash), it may be feasible to extract alumina from ARCG using hydro-chemical process. However, the research on the extraction of alumina from ARCG by hydro-chemical process has not been reported. The aim of this study is to extract alumina from ARCG by the hydro-chemical process and determine the behaviour of $Al_2O_3$ during the extraction process. Meanwhile, the phase transformation of ARCG in hydro-chemical process is also researched.

# 2. Experimental

## 2.1. Raw materials

The ARCG was derived from a coal mine of Shanxi Province, China. Samples of ARCG were crushed to less than 2.36 mm and dried in an oven at 105°C for 24 h. Then, the representative sample was obtained by the cone and quartering method.

The reagents NaOH, $Ca(OH)_2$ and $Al(OH)_3$ used in the experiment were of analytical purity grade offered by Sinopharm Chemical Reagents Co., Ltd (Shanghai, China) and used without further purification. Ultrapure water was obtained from a Milli-Q water purification system (Millipore, USA).

## 2.2. Experimental apparatus and procedure

A 500 ml sealed nickel-lined stainless steel autoclave equipped with a water cooling system, a mechanical agitator and an external electro-thermal furnace was employed to extract alumina from ARCG. The heating and agitation rate of the autoclave were controlled by using an automatic control system to maintain a desired temperature and agitation speed. Extraction of alumina experiments were performed in the autoclave. For each experiment, 20 g of ARCG, 14.5 g of $Ca(OH)_2$ and a certain proportion of sodium aluminate solution with a caustic ratio (molar ratio of $Na_2O$ to $Al_2O_3$) of 25 were mixed. Then, the alumina extraction reaction can take place under certain temperatures and pressures. After the extraction reaction, the reactor was cooled down to ambient temperature using cooling water. Then, the separation of solid and liquid was achieved by the filtration process. Finally, the solid phase was washed three times with deionized water and then dried in an oven at 105°C for 12 h prior to analyses. Each experiment was repeated at least three times. All data are expressed as mean ± s.d.

## 2.3. Analysis methods

The phase structures were characterized by X-ray diffraction (X'Pert powder, PANalytical, The Netherlands, 40 kV, 30 mA, Cu K$\alpha$ as X-ray source). Inductively coupled plasma-optical emission spectrometry (ICP-OES, PE Optima 5300DV, Perkin-Elmer) was used for analysing the chemical composition of both liquid and solid samples. Surface morphology was examined by a scanning electron microscope (S-4800, Hitachi, Japan). Chemical components of different regions in the samples were determined by energy-dispersive X-ray spectroscopy (EDS).

The alumina extraction rate ($\eta_A$) was calculated by following formula:

$$\eta_A = \frac{v \times c}{m \times w} \times 100\%,$$

where $v$ is the volume of leaching liquor (l), $c$ is the concentration of $Al_2O_3$ in leaching liquor (g l$^{-1}$), $m$ is the mass of ARCG (g) and $w$ is the content of $Al_2O_3$ in ARCG (g g$^{-1}$).

# 3. Results and discussion

## 3.1. Characterization of the raw materials

The chemical analysis of the ARCG is shown in table 1. As seen from the table 1, the ARCG is composed chiefly of $Al_2O_3$, $SiO_2$, $Fe_2O_3$, CaO and $TiO_2$, and the content of alumina and silica are 44.93% and 54.13%, respectively. The mass ratio of alumina to silica (Al/Si) is 0.83, more close to the theoretical value of 0.85 for kaolinite ($Al_2O_3.2SiO_2.2H_2O$). According to the XRD patterns shown in figure 1, the major phases of ARCG are quartz ($SiO_2$) and kaolinite ($Al_2O_3.2SiO_2.2H_2O$). Figure 1 also reveals that element of Al in ARCG is dominated by kaolinite phase, as there exist no other alumina-bearing crystalline phase. The morphologies of the ARCG are presented in figure 2, showing an irregular shape with loose and flaky structure. Figure 2 indicates that the particle size is small (about 200 nm).

## 3.2. Effect of NaOH concentration

The effects of NaOH concentration on the alumina extraction rate are displayed in figure 3. From figure 3, it is obviously observed that the alumina extraction ratio increased with the NaOH concentration increased from 40.0% to 47.5% and then decreased with further increase of NaOH concentration.

Figure 4 illustrates the XRD patterns within the NaOH concentration ranging from 40.0% to 50.0%. When the NaOH concentration was 40.0%, the residue phases were mainly $Na_8Al_6Si_6O_{24}(OH)_2(H_2O)_2$, $Ca_2Al_2SiO_6(OH)_2$, $Ca(OH)_2$ and $NaCaHSiO_4$ with weak peaks. The peaks intensity of $Na_8Al_6Si_6O_{24}(OH)_2(H_2O)_2$ and $Ca_2Al_2SiO_6(OH)_2$ decreased gradually with increased NaOH concentration, while peaks intensity of $NaCaHSiO_4$ showed an opposite tendency. When the NaOH concentration reached 47.5%, the characteristic peaks of $Na_8Al_6Si_6O_{24}(OH)_2(H_2O)_2$ and $Ca_2Al_2SiO_6(OH)_2$ completely disappeared and $NaCaHSiO_4$ with strong peaks formed, resulting in high alumina extraction ratio. When the NaOH concentration reached 50.0%, the $NaCaHSiO_4$ peaks got weaker and peaks of $1.2Na_2O \cdot 0.8CaO \cdot Al_2O_3 \cdot 2SiO_2 \cdot H_2O$ emerged. The generation of $Na_8Al_6Si_6O_{24}(OH)_2(H_2O)_2$, $Ca_2Al_2SiO_6(OH)_2$ and $1.2Na_2O \cdot 0.8CaO \cdot Al_2O_3 \cdot 2SiO_2 \cdot H_2O$ was harmful to alumina extraction reaction. This can be the explanation of the lower alumina extraction ratio when the concentration of NaOH was too high or too low. It should be

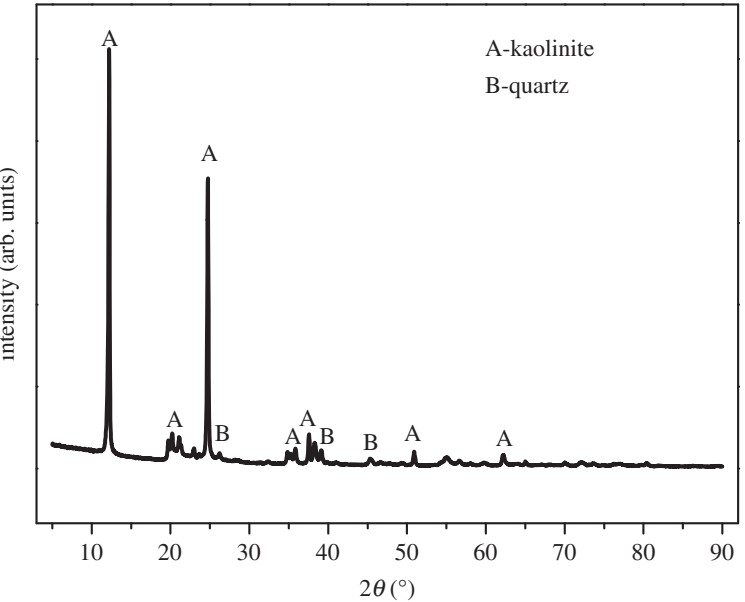

**Figure 1.** XRD patterns of the ARCG.

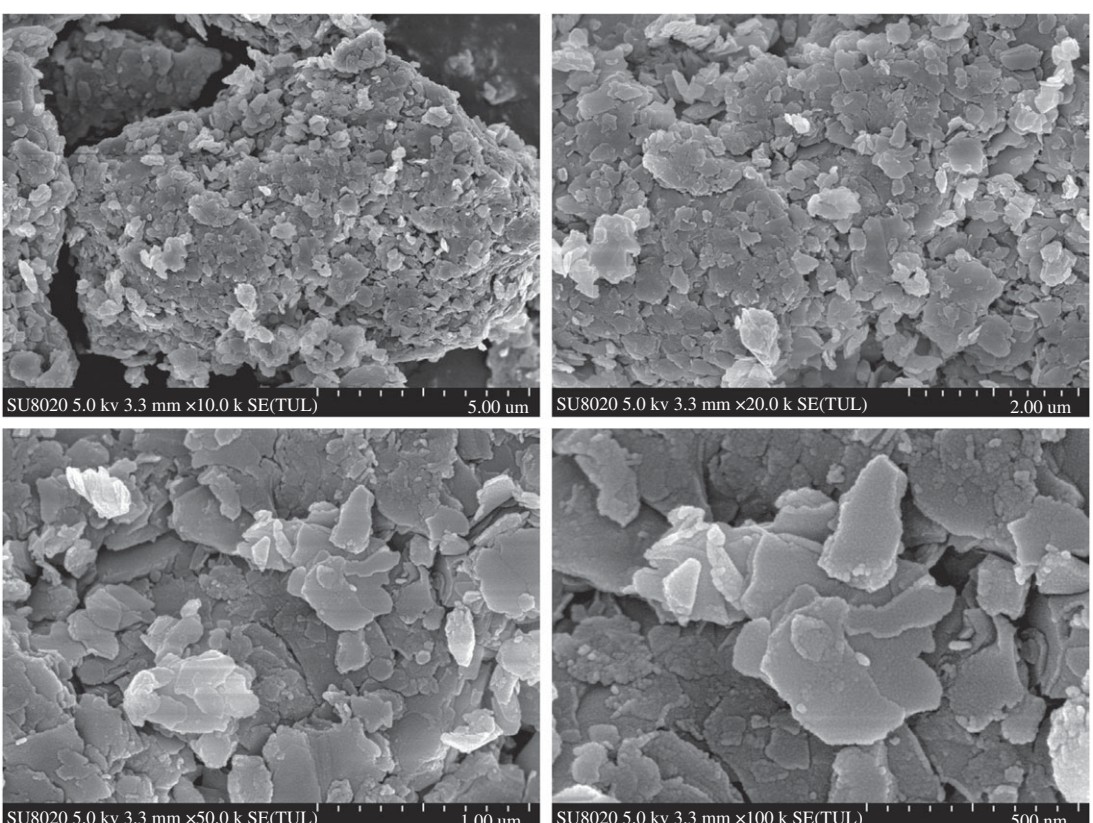

**Figure 2.** SEM images of ARCG.

**Table 1.** Chemical composition of the ARCG. Al/Si is the $Al_2O_3$-to-$SiO_2$ mass ratio.

| composition | $Al_2O_3$ | $SiO_2$ | $Fe_2O_3$ | CaO | $TiO_2$ | $Na_2O$ | Al/Si |
|---|---|---|---|---|---|---|---|
| content (wt%) | 44.93 | 54.13 | 0.18 | 0.09 | 0.42 | 0.02 | 0.83 |

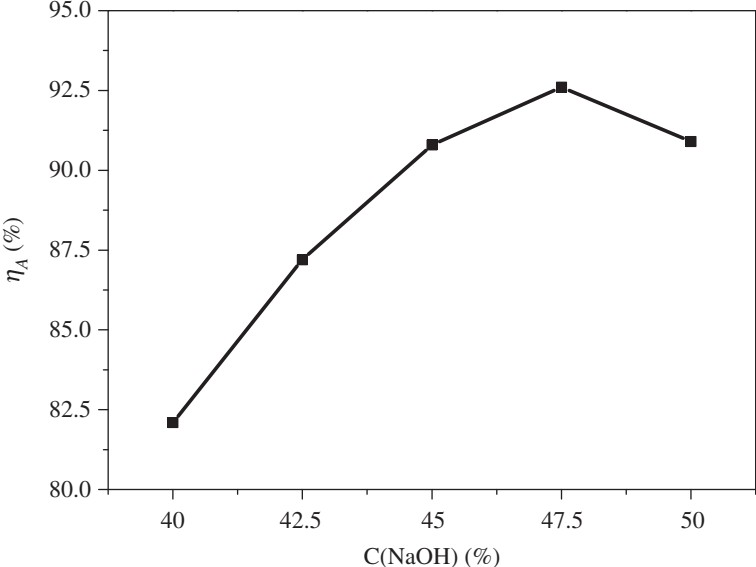

**Figure 3.** Effects of alkali concentration on extraction rate of Al$_2$O$_3$.

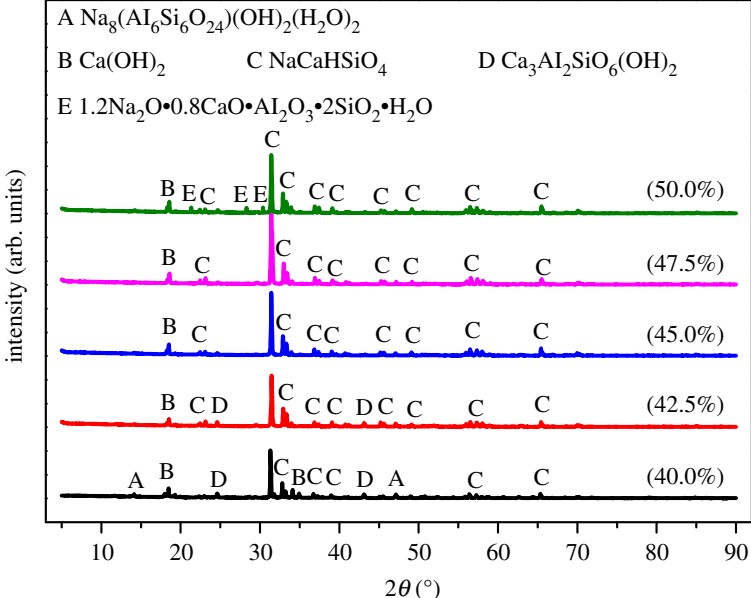

**Figure 4.** XRD patterns of ARCG residue obtained at different alkali concentration.

noted that Ding *et al*. [20] also revealed the existence of 1.2Na$_2$O $\cdot$ 0.8CaO $\cdot$ Al$_2$O$_3$ $\cdot$ 2SiO$_2$ $\cdot$ H$_2$O during a similar operation in treating fly ash. The study believes that formation of 1.2Na$_2$O $\cdot$ 0.8CaO $\cdot$ Al$_2$O$_3$ $\cdot$ 2SiO$_2$ $\cdot$ H$_2$O can be avoided by increasing stirring speed to above 650 r.p.m. However, this study found that high alkaline concentration was favourable to the formation of 1.2Na$_2$O $\cdot$ 0.8CaO $\cdot$ Al$_2$O$_3$ $\cdot$ 2SiO$_2$ $\cdot$ H$_2$O.

## 3.3. Effect of alkali to gangue ratio

The alkali to gangue ratio usually has an important influence on the design of reaction vessel and circulation quantity of reaction medium. As shown in figure 5, the alumina extraction ratio increased monotonically as alkali to gangue ratio increased. However, when the alkali to gangue ratio was more than 6, the extraction ratio changed slightly. Hence, alkali to gangue ratio of 6 was selected for other experiments as one optimal variable.

Figure 6 shows the XRD patterns with various alkali to gangue ratios. When the ratio was of 5, the slag phases were mainly Na$_8$Al$_6$Si$_6$O$_{24}$(OH)$_2$(H$_2$O)$_2$, Ca$_2$Al$_2$SiO$_6$(OH)$_2$, Ca(OH)$_2$ and NaCaHSiO$_4$. With

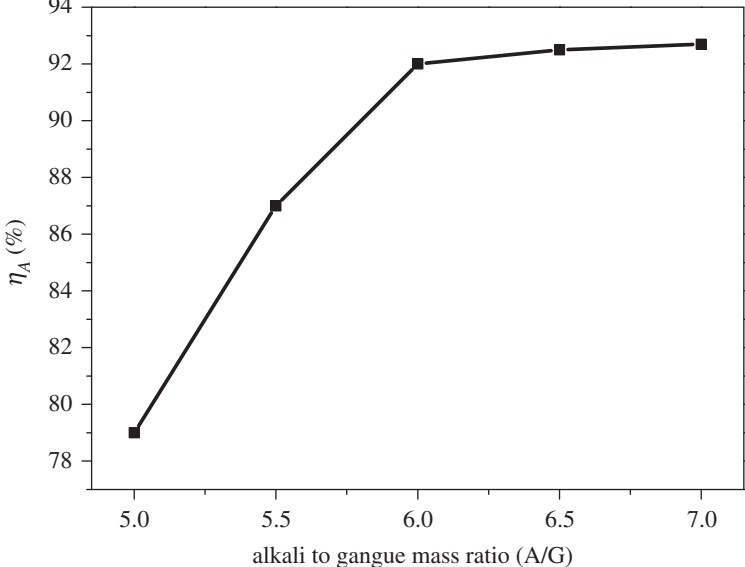

**Figure 5.** Effects of alkali to gangue ratio on extraction rate of $Al_2O_3$.

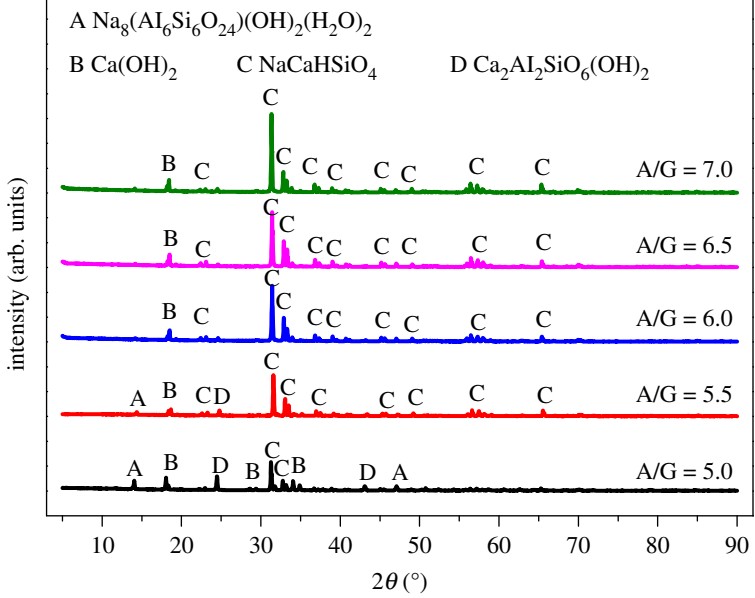

**Figure 6.** XRD patterns of ARCG residue obtained at different alkali to gangue ratio.

the rising of alkali to gangue ratio, $Na_8Al_6Si_6O_{24}(OH)_2(H_2O)_2$ and $Ca_2Al_2SiO_6(OH)_2$ peaks got weaker and peaks of $NaCaHSiO_4$ became stronger, which can explain the reason of better alumina extraction effect at higher alkali to gangue ratios.

## 3.4. Effect of reaction temperature and time

Reaction temperature is closely related to energy consumption of the process. From figure 7, it is observed that the extraction ratio of $Al_2O_3$ increased significantly as the reaction temperature rose from 200°C to 260°C and changed slightly as the temperature increased from 260°C to 280°C. Hence, the optimal temperature for extraction of alumina was selected as 260°C.

Figure 8 shows the phase transformation as the temperature rising. When the reaction temperature was at 200°C, the phases of the residue had transformed. Characteristic peaks of kaolinite existing in XRD patterns of ARCG disappeared, and new peaks of $Na_8Al_6Si_6O_{24}(OH)_2(H_2O)_2$, $Ca_2Al_2SiO_6(OH)_2$, $Ca(OH)_2$ and $NaCaHSiO_4$ emerged in the leached residue. When the reaction temperature reached 220°C, the peaks intensity of $Na_8Al_6Si_6O_{24}(OH)_2(H_2O)_2$ and $Ca_2Al_2SiO_6(OH)_2$ had an increasing

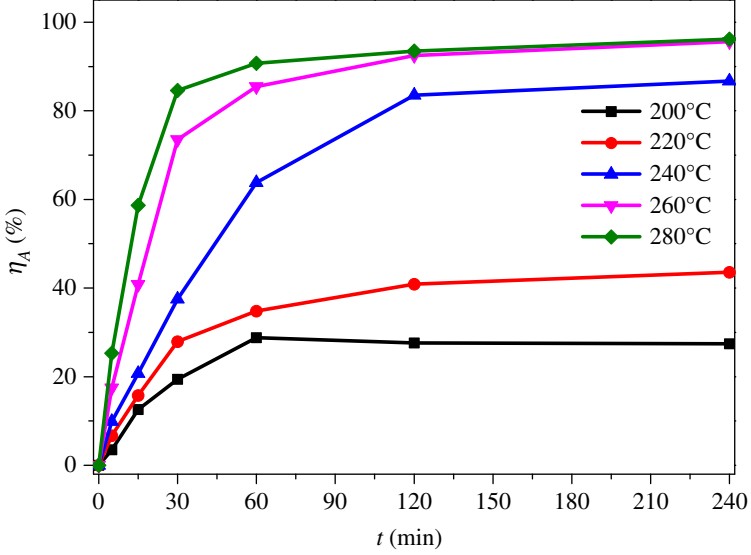

**Figure 7.** Effects of reaction temperature on extraction rate of $Al_2O_3$.

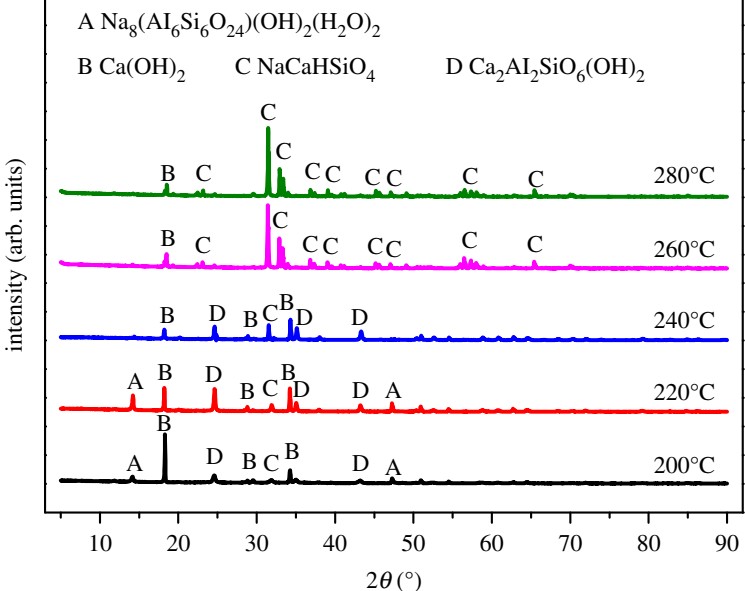

**Figure 8.** XRD patterns of ARCG residue obtained at different temperatures.

tendency, which was similar to the previous research in treating fly ash [31,33]. When the temperature increased from 220°C to 260°C, the peaks of $Na_8Al_6Si_6O_{24}(OH)_2(H_2O)_2$ and $Ca_2Al_2SiO_6(OH)_2$ became weaker and disappeared. Finally, the $NaCaHSiO_4$ became the only stable phase, corresponding to a better alumina extraction effect.

Reaction time is another key technical index on extraction of alumina from ARCG. From figure 7, it is obviously observed that the alumina extraction ratio increased gradually with the increase of reaction time. However, when the reaction time was over 120 min, the extraction ratio went near to steadiness. Hence, the optimal reaction time was determined as 120 min.

The phase change of leached residues at different leaching time is shown in figure 9. It can be seen from figure 9 that when the reaction time was 5 min, the kaolinite characteristic peak of coal gangue had disappeared, and peaks of $Na_8Al_6Si_6O_{24}(OH)_2(H_2O)_2$, $Ca_2Al_2SiO_6(OH)_2$, $NaCaHSiO_4$ and $Ca(OH)_2$ were detected. With the extension of the reaction time to 15 min, the characteristic peaks of $Na_8Al_6Si_6O_{24}(OH)_2(H_2O)_2$ turn weak, and $Ca_2Al_2SiO_6(OH)_2$ and $NaCaHSiO_4$ became stronger. As the reaction time continued to extend, the characteristic peaks of $Na_8Al_6Si_6O_{24}(OH)_2(H_2O)_2$ and $Ca_2Al_2SiO_6(OH)_2$ disappeared gradually, and peaks of $NaCaHSiO_4$ enhanced obviously.

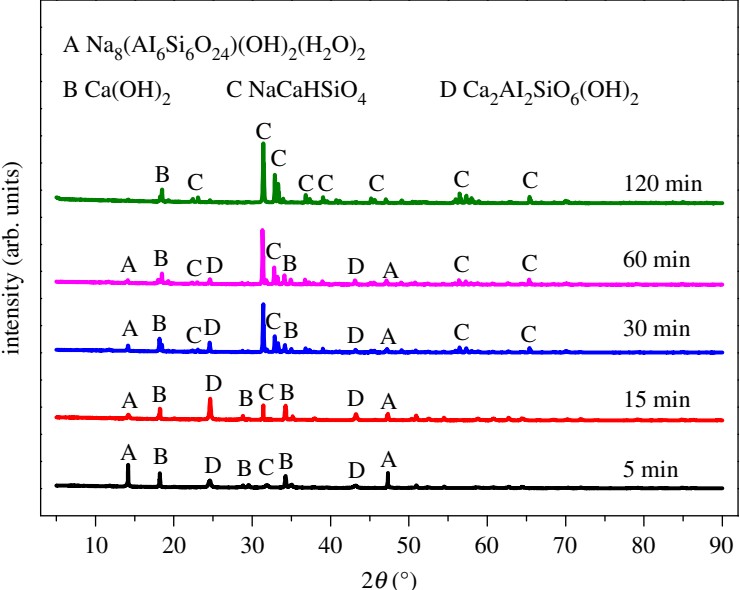

**Figure 9.** XRD patterns of ARCG residue obtained at different reaction time.

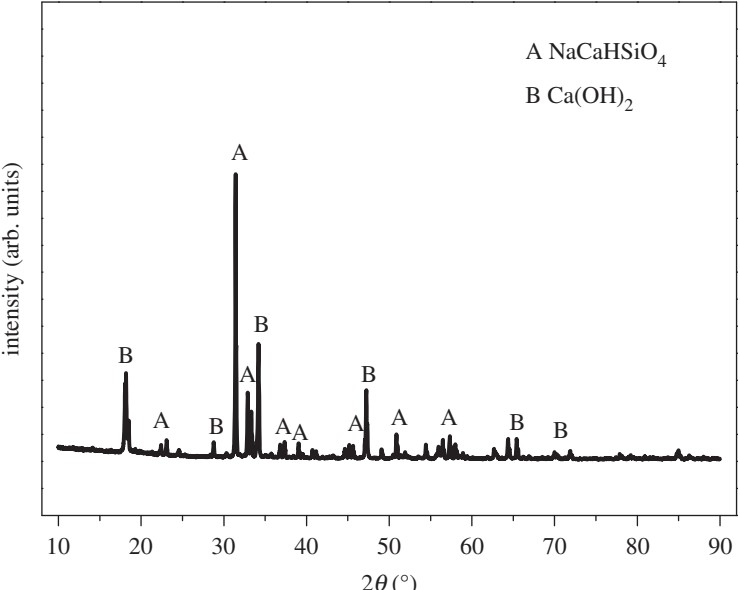

**Figure 10.** XRD patterns of ARCG residue obtained under optimum conditions.

**Table 2.** Composition of ARCG residue obtained under optimized conditions. Al/Si is the $Al_2O_3$-to-$SiO_2$ mass ratio.

| composition | $Al_2O_3$ | $SiO_2$ | CaO | $Na_2O$ | Al/Si |
|---|---|---|---|---|---|
| content (wt%) | 1.64 | 37.63 | 44.01 | 19.28 | 0.04 |

## 3.5. Optimum conditions

From the above-mentioned experiments, the results show that the optimum conditions to extract alumina from ARCG are: NaOH concentration of 47.5%, alkali to gangue ratio of 6, reaction temperature of 260°C and reaction time of 120 min. Under the optimum conditions, alumina of ARCG can be extracted efficiently. The chemical composition of leached residue under optimized conditions is shown in table 2, the $Al_2O_3$ content and alumina to silica ratio decreased dramatically compared with the

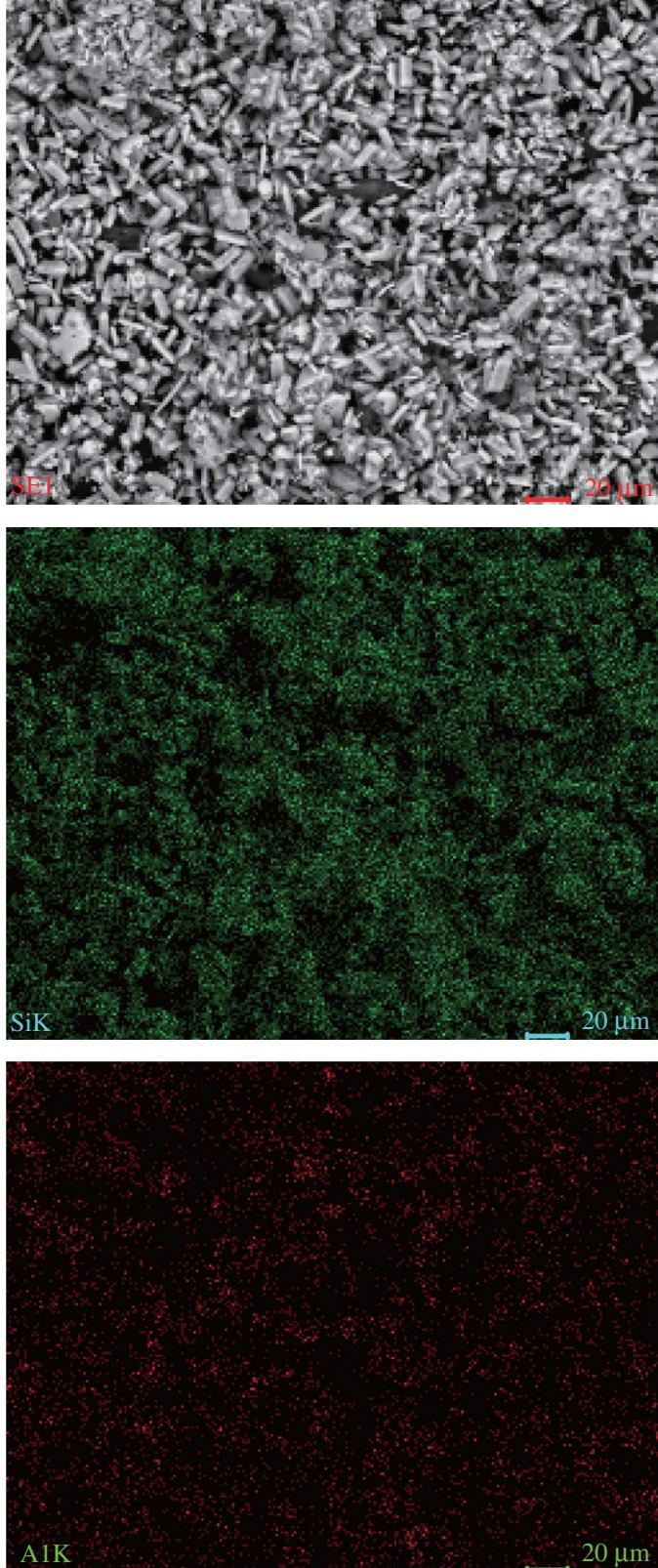

**Figure 11.** SEM and EDS mapping images of ARCG residue obtained under optimum conditions.

untreated ARCG and fell to 1.64% and 0.04, respectively. Meanwhile, the content of calcium and sodium increased significantly. The XRD patterns are shown in figure 10, confirming that the major phase of leached residue is $NaCaHSiO_4$, which demonstrates that separation of Al and Si in ARCG can be realized by hydro-chemical treatment. Micrographs and element distribution analysis results are

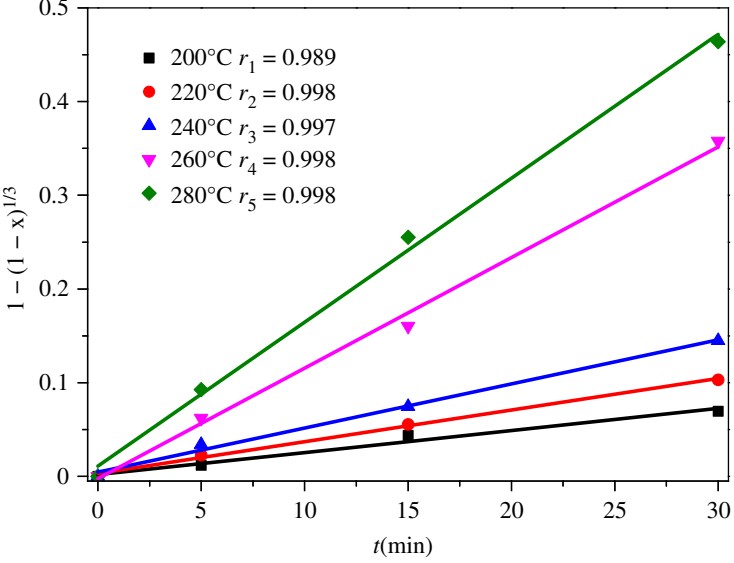

**Figure 12.** Plot of $1 - (1 - x)^{1/3}$ versus time.

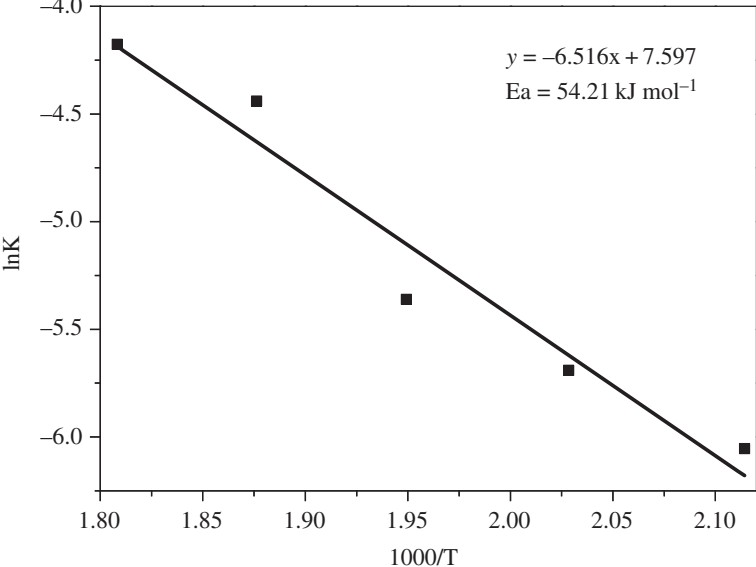

**Figure 13.** Arrhenius plot for alumina extraction during 200℃ to 280℃.

shown in figure 11. As shown in figure 11, the particle shapes changed markedly compared with untreated ARCG and transformed from flaky shape to long columnar shape. The results from surface scanning analysis confirmed that the element of Si was distributed intensively and mainly concentrated in the columnar-like particles. However, the distribution of Al was much less compared with Si element, indicating a lower content of Al. As the alumina-free phase $NaCaHSiO_4$ is dominant in leached residue, so the slight amounts of Al existing in the residue may be caused by adsorption.

Based on the above discussion, alumina-containing phases $Na_8Al_6Si_6O_{24}(OH)_2(H_2O)_2$ and $Ca_2Al_2SiO_6(OH)_2$ can be formed at relatively low temperature and short reaction time via reaction equations (3.1) and (3.2). To obtain high alumina extraction rate from ARCG, the target product is $NaCaHSiO_4$, which is generated by reaction equations (3.3) and (3.4) under the optimum conditions.

$$3(Al_2O_3 \cdot 2SiO_2 \cdot 2H_2O) + 8NaOH = Na_8Al_6Si_6O_{24}(OH)_2(H_2O)_2 + 7H_2O, \tag{3.1}$$

$$Al_2O_3 \cdot 2SiO_2 \cdot 2H_2O + 3Ca(OH)_2 + NaOH = Ca_2Al_2SiO_6(OH)_2 + NaCaHSiO_4 + 4H_2O, \tag{3.2}$$

$$Na_8Al_6Si_6O_{24}(OH)_2(H_2O)_2 + 6Ca(OH)_2 + 4NaOH = 6NaCaHSiO_4 + 6NaAlO_2 + 8H_2O \tag{3.3}$$

$$\text{and} \quad Al_2O_3 \cdot 2SiO_2 \cdot 2H_2O + 4NaOH + 2Ca(OH)_2 = 2NaAlO_2 + 2NaCaHSiO_4 + 5H_2O. \tag{3.4}$$

Under this optimal condition, the generation of $Na_8Al_6Si_6O_{24}(OH)_2(H_2O)_2$ and $Ca_2Al_2SiO_6(OH)_2$ can be obviously suppressed; $SiO_2$ in ARCG can be fixed in the form of $NaCaHSiO_4$, so that alumina loss can be avoided theoretically by hydro-chemical process.

## 3.6 Kinetic of alumina extraction

The fitting equation based on interfacial diffusion control has successfully been applied to describe the leaching dynamics characteristics of various extraction reactions [34,35]. In this thesis, the equation was also used to analyse the kinetics behaviour of extraction alumina from ARCG. As reported in figure 7, the experimental data obtained at 200, 220, 240, 260 and 280°C were used for kinetic analysis. The experimental data can be fitted by $1 - (1 - X)^{1/3} = kt$ perfectly, as shown in figure 12, where X is the extraction rate of alumina, t is the reaction time and k is the rate constant. The apparent activation energy of the alumina extraction process was determined based on Arrhenius equation representing the relevance of the natural logarithm of reaction rate versus $1/T$, as shown in figure 13. The apparent activation energy obtained from the slope of the straight line was $54.21 \, \text{kJ mol}^{-1}$.

# 4. Conclusion

A novel process for alumina extraction from ARCG was investigated systematically. The optimum conditions are: NaOH concentration of 47.5%, alkali to gangue ratio of 6, reaction temperature of 260°C and reaction time of 120 min. Under these conditions, the alumina to silica ratio of residue and alumina extraction rate achieved 0.04 and 94.68%, respectively. $NaCaHSiO_4$ was identified as the major phase of leached residue, which contributing to high effective extraction of alumina. The kinetics of extraction of alumina from ARCG has also been studied at temperatures of 200, 220, 240 260 and 280°C, and the apparent activation energy was calculated as $54.21 \, \text{kJ mol}^{-1}$. The study indicates that the hydro-chemical process had good potential industry application prospect for extraction of alumina from ARCG.

Data accessibility. The datasets supporting this article are provided in the electronic supplementary material.

Authors' contributions. Q.Y. designed the experimental process, participated in data analysis and wrote the manuscript. F.Z. and X.D. performed the experimental work. H.G. and C.Z. carried out the sample analysis. C.S. and M.Z coordinated the study and participated in data analysis. All authors gave final approval for publication.

Competing interests. We declare we have no competing interest.

Funding. This research was supported by the Natural Science Foundation of Hebei Province (E2018508105), the National Natural Science Foundation of China (51674119), Fundamental Research Funds for the Central Universities (3142017010 and 3142017104) and Program for the Top Young Talents of Higher Learning Institutions of Hebei Province (BJ2018201).

Acknowledgements. We would like to express appreciation to the staff of Analysis and Measurement Center of Chinese Academy of Sciences. Additionally, we would also like to thank the anonymous reviewers for their helpful comments.

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
