## [Reviewer comments · Royal Society Open Science]

Review History

RSOS-192132.R0 (Original submission)

Review form: Reviewer 1

Is the manuscript scientifically sound in its present form?

No

Are the interpretations and conclusions justified by the results?

No

Is the language acceptable?

Yes

Do you have any ethical concerns with this paper?

No

Have you any concerns about statistical analyses in this paper?

Yes

Recommendation?

Major revision is needed (please make suggestions in comments)

Comments to the Author(s)

This article reports the extraction of alumina from alumina rich coal gangue by a hydro-chemical process. The influences of NaOH concentration, mass ratio of alkali to gangue, reaction temperature, and residence time were systematically studied.

Though some characterizations and analysis to verify the conclusions, but there are some details should be checked before its publication.

1. In Figure 2, there is the SEM image of only one sample, without comparison, is not enough to show that a small particle size is conducive to the extraction reaction? Or are there any other references?
2. In the analysis of the effect of NaOH concentration on the extraction rate of aluminum, when the NaOH concentration was of 40%, there are four residue phases, but only three are marked in Figure 4, the $\text{Ca}_2\text{Al}_2\text{SiO}_6(\text{OH})_2$ is not marked.
3. In Figure 3, why the concentration rate at NaOH concentration of 50% is lower than that of the NaOH concentration of 42.5%, but in Figure 4, the peak strength of NaCaHSiO_4 in the residue phase with NaOH concentration of 50% is much higher?
4. In the analysis of the effect of alkali to gangue ratio on the extraction rate of aluminum, describes that with the alkali to gangue ratio rising, $\text{Na}_8\text{Al}_6\text{Si}_6\text{O}_{24}(\text{OH})_2(\text{H}_2\text{O})_2$ and $\text{Ca}_2\text{Al}_2\text{SiO}_6(\text{OH})_2$ peaks became weaken and peaks of NaCaHSiO_4 became stronger. But why is the peak strength of the $\text{Na}_8\text{Al}_6\text{Si}_6\text{O}_{24}(\text{OH})_2(\text{H}_2\text{O})_2$ and $\text{Ca}_2\text{Al}_2\text{SiO}_6(\text{OH})_2$ phases at a ratio of 6 is significantly higher than the peak at the ratio is of 5?
5. In XRD analysis, as the reaction temperature rising from 200 to 260°C. The peaks of $\text{Na}_8\text{Al}_6\text{Si}_6\text{O}_{24}(\text{OH})_2(\text{H}_2\text{O})_2$ and $\text{Ca}_2\text{Al}_2\text{SiO}_6(\text{OH})_2$ became weaker, But why is the peak strength of the $\text{Na}_8\text{Al}_6\text{Si}_6\text{O}_{24}(\text{OH})_2(\text{H}_2\text{O})_2$ and $\text{Ca}_2\text{Al}_2\text{SiO}_6(\text{OH})_2$ phases at 220°C higher than the peak when the reaction temperature is at 200°C?
6. In Figure 7, No experiment was performed with a reaction temperature of 280°C or higher temperature. Why can it be stated that 260°C is the optimal reaction temperature?
7. In figure 11, the SEM image is not clear enough.
8. Why Table 1 is represented by Al/Si and Table 2 is represented by A/S?
9. There are a lot of annotations, formatting, and language errors in the manuscript. For examples, in Figure 6、Figure 8 and Figure 9, the author write NaCaHSiO_4 as NaCHSiO_4 . The author needs to carefully review and improve the manuscript.

Review form: Reviewer 2

Is the manuscript scientifically sound in its present form?

No

Are the interpretations and conclusions justified by the results?

Yes

Is the language acceptable?

Yes

Do you have any ethical concerns with this paper?

No

Have you any concerns about statistical analyses in this paper?

No

Recommendation?

Reject

Comments to the Author(s)

This paper investigated the extraction of alumina from alumina rich coal gangue by a hydro-chemical process. However, it is believed that this article didn't give some valuable scientific discussion about the alumina extraction process. The organization and writing of article were extremely similar with the ref. [31], their previous research paper with similar hydro-chemical process to fly ash. Therefore, It's not suitable for the publication of the research article type.

Review form: Reviewer 3 (Wei Xiao)**Is the manuscript scientifically sound in its present form?**

Yes

Are the interpretations and conclusions justified by the results?

No

Is the language acceptable?

Yes

Do you have any ethical concerns with this paper?

No

Have you any concerns about statistical analyses in this paper?

No

Recommendation?

Major revision is needed (please make suggestions in comments)

Comments to the Author(s)

It is interesting that the processing of extracting alumina from ARCG was studied in this paper. The paper found that all of the NaOH concentration, mass ratio of alkali to gangue, reaction temperature, and residence time could affect the alumina extraction rate. I think their study is timely and merits publication in principle. However, major revisions are required before it might be published in RSOS. I therefore encourage the authors to revise the following outstanding points.

1. The ARCG sample selected in this paper comes from a coal mine in Shanxi Province, but according to the chemical analysis and XRD results of the sample, the main component of the ARCG sample is kaolin, which seems to deviate from the theme of this paper. It is better to extract aluminum from kaolin than from ARCG.
2. The purity and grade of reagents and water should be indicated in the experiment. P7 line 4.
3. In P7 line 36, does the content of Al₂O₃ refer to Al₂O₃ in ARCG? If so, then the content of Al₂O₃ is fixed. According to the fixed value of Al₂O₃/NaOH, the amount of NaOH added in each experiment should be fixed.
4. In P6 line 58, the unit of granularity should be μ m or mm.
5. In the experiment of Al₂O₃ extraction, the average value should be taken from the parallel experiment of multiple units, and the error range should be explained.
6. In Section 2.3, the experimental process of XRD is not clear, so the XRD analysis method in the reference "Comparative studies on catalytic mechanisms for natural chalcopyrite-induced Fenton oxidation: Effect of chalcopyrite type. Journal of Hazardous Materials 2020, 381, 120998" can be used.

Decision letter (RSOS-192132.R0)

30-Jan-2020

Dear Dr Yang:

Title: Extraction of alumina from alumina rich coal gangue by a hydro-chemical process
Manuscript ID: RSOS-192132

The editor assigned to your manuscript has now received comments from reviewers. We would like you to revise your paper in accordance with the referee and Subject Editor suggestions which can be found below (not including confidential reports to the Editor). Please note this decision does not guarantee eventual acceptance.

Please submit your revised paper before 22-Feb-2020. Please note that the revision deadline will expire at 00.00am on this date. If we do not hear from you within this time then it will be assumed that the paper has been withdrawn. In exceptional circumstances, extensions may be possible if agreed with the Editorial Office in advance. We do not allow multiple rounds of revision so we urge you to make every effort to fully address all of the comments at this stage. If deemed necessary by the Editors, your manuscript will be sent back to one or more of the original reviewers for assessment. If the original reviewers are not available we may invite new reviewers.

RSC Associate Editor:

Comments to the Author:

Please make it clear how the work presented differs from what has been previously published.

RSC Subject Editor:

Comments to the Author:

(There are no comments.)

Reviewers' Comments to Author:

Reviewer: 1

Comments to the Author(s)

This article reports the extraction of alumina from alumina rich coal gangue by a hydro-chemical process. The influences of NaOH concentration, mass ratio of alkali to gangue, reaction temperature, and residence time were systematically studied.

Though some characterizations and analysis to verify the conclusions, but there are some details should be checked before its publication.

1. In Figure 2, there is the SEM image of only one sample, without comparison, is not enough to show that a small particle size is conducive to the extraction reaction? Or are there any other references?
2. In the analysis of the effect of NaOH concentration on the extraction rate of aluminum, when the NaOH concentration was of 40%, there are four residue phases, but only three are marked in Figure 4, the $\text{Ca}_2\text{Al}_2\text{SiO}_6(\text{OH})_2$ is not marked.
3. In Figure 3, why the concentration rate at NaOH concentration of 50% is lower than that of the NaOH concentration of 42.5%, but in Figure 4, the peak strength of NaCaHSiO_4 in the residue phase with NaOH concentration of 50% is much higher?
4. In the analysis of the effect of alkali to gangue ratio on the extraction rate of aluminum, describes that with the alkali to gangue ratio rising, $\text{Na}_8\text{Al}_6\text{Si}_6\text{O}_{24}(\text{OH})_2(\text{H}_2\text{O})_2$ and $\text{Ca}_2\text{Al}_2\text{SiO}_6(\text{OH})_2$ peaks became weaken and peaks of NaCaHSiO_4 became stronger. But why is the peak strength of the $\text{Na}_8\text{Al}_6\text{Si}_6\text{O}_{24}(\text{OH})_2(\text{H}_2\text{O})_2$ and $\text{Ca}_2\text{Al}_2\text{SiO}_6(\text{OH})_2$ phases at a ratio of 6 is significantly higher than the peak at the ratio is of 5?
5. In XRD analysis, as the reaction temperature rising from 200 to 260°C. The peaks of $\text{Na}_8\text{Al}_6\text{Si}_6\text{O}_{24}(\text{OH})_2(\text{H}_2\text{O})_2$ and $\text{Ca}_2\text{Al}_2\text{SiO}_6(\text{OH})_2$ became weaker, But why is the peak strength of the $\text{Na}_8\text{Al}_6\text{Si}_6\text{O}_{24}(\text{OH})_2(\text{H}_2\text{O})_2$ and $\text{Ca}_2\text{Al}_2\text{SiO}_6(\text{OH})_2$ phases at 220°C higher than the peak when the reaction temperature is at 200°C?
6. In Figure 7, No experiment was performed with a reaction temperature of 280°C or higher temperature. Why can it be stated that 260°C is the optimal reaction temperature?
7. In figure 11, the SEM image is not clear enough.
8. Why Table 1 is represented by Al/Si and Table 2 is represented by A/S?
9. There are a lot of annotations, formatting, and language errors in the manuscript. For examples, in Figure 6、Figure 8 and Figure 9, the author write NaCaHSiO_4 as NaCHSiO_4 . The author needs to carefully review and improve the manuscript.

Reviewer: 2

Comments to the Author(s)

This paper investigated the extraction of alumina from alumina rich coal gangue by a hydro-chemical process. However, it is believed that this article didn't give some valuable scientific discussion about the alumina extraction process. The organization and writing of article were extremely similar with the ref. [31], their previous research paper with similar hydro-chemical process to fly ash. Therefore, It's not suitable for the publication of the research article type.

Reviewer: 3

Comments to the Author(s)

It is interesting that the processing of extracting alumina from ARCG was studied in this paper. The paper found that all of the NaOH concentration, mass ratio of alkali to gangue, reaction temperature, and residence time could affect the alumina extraction rate. I think their study is timely and merits publication in principle. However, major revisions are required before it might be published in RSOS. I therefore encourage the authors to revise the following outstanding points.

1. The ARCG sample selected in this paper comes from a coal mine in Shanxi Province, but according to the chemical analysis and XRD results of the sample, the main component of the ARCG sample is kaolin, which seems to deviate from the theme of this paper. It is better to extract aluminum from kaolin than from ARCG.
2. The purity and grade of reagents and water should be indicated in the experiment. P7 line 4.
3. In P7 line 36, does the content of Al₂O₃ refer to Al₂O₃ in ARCG? If so, then the content of Al₂O₃ is fixed. According to the fixed value of Al₂O₃/NaOH, the amount of NaOH added in each experiment should be fixed.
4. In P6 line 58, the unit of granularity should be μ m or mm.
5. In the experiment of Al₂O₃ extraction, the average value should be taken from the parallel experiment of multiple units, and the error range should be explained.
6. In Section 2.3, the experimental process of XRD is not clear, so the XRD analysis method in the reference "Comparative studies on catalytic mechanisms for natural chalcopyrite-induced Fenton oxidation: Effect of chalcopyrite type. Journal of Hazardous Materials 2020, 381, 120998" can be used.

Author's Response to Decision Letter for (RSOS-192132.R0)

See Appendix A.

RSOS-192132.R1 (Revision)

Review form: Reviewer 1

Is the manuscript scientifically sound in its present form?

Yes

Are the interpretations and conclusions justified by the results?

Yes

Is the language acceptable?

Yes

Do you have any ethical concerns with this paper?

No

Have you any concerns about statistical analyses in this paper?

No

Recommendation?

Accept as is

Comments to the Author(s)

The authors have revised the manuscript following all my suggestions, I agree for its publication in the present form

Decision letter (RSOS-192132.R1)

26-Mar-2020

Dear Dr Yang:

Title: Extraction of alumina from alumina rich coal gangue by a hydro-chemical process
Manuscript ID: RSOS-192132.R1

It is a pleasure to accept your manuscript in its current form for publication in Royal Society Open Science. The chemistry content of Royal Society Open Science is published in collaboration with the Royal Society of Chemistry.

RSC Associate Editor:
Comments to the Author:
(There are no comments.)

RSC Subject Editor:
Comments to the Author:
(There are no comments.)

Reviewer(s)' Comments to Author:

Reviewer: 1

Comments to the Author(s)

The authors have revised the manuscript following all my suggestions, I agree for its publication in the present form

Appendix A

Response to the referees' comments:

Dear referees:

Re: Manuscript ID: RSOS-192132

Manuscript Title: Extraction of alumina from alumina rich coal gangue by a hydro-chemical process

Authors: Quancheng Yang, Fan Zhang, Xingjian Deng, Hongchen Guo, Chao Zhang, Changsheng Shi, Ming Zeng

We would like to thank both the editor and the referees for comments concerning our manuscript entitled “Extraction of alumina from alumina rich coal gangue by a hydro-chemical process” (ID: RSOS-192132). Those comments are very helpful to improve our manuscript. The responses to the reviewer’s comments are listed as following:

Best regards.

Sincerely yours,

Dr Quancheng Yang

Department of Environmental Engineering, North China Institute of Science and Technology, Beijing 101601, P.R. China

Email: yangquancheng@126.com

February 21th, 2020

Reviewer 1:

Comment 1: In Figure 2, there is the SEM image of only one sample, without comparison, is not enough to show that a small particle size is conducive to the extraction reaction? Or are there any other references?.

Response: Thank you for your valuable comments. We are sorry for the unclear description of this sentence. Based on the reviewers' comment, we have removed the words “which is conducive to the extraction reaction” in the revised manuscript to make the expression more rigorous.

Comment 2: In the analysis of the effect of NaOH concentration on the extraction rate of aluminum, when the NaOH concentration was of 40%, there are four residue phases, but only three are marked in Figure 4, the $\text{Ca}_2\text{Al}_2\text{SiO}_6(\text{OH})_2$ is not marked.

Response: Thank you very much to point out the faults in our manuscript. We are very sorry for misusing an unfinished XRD diagram in Figure 4. We have corrected it in the revised manuscript.

Comment 3: In Figure 3, why the concentration rate at NaOH concentration of 50% is lower than that of the NaOH concentration of 42.5%, but in Figure 4, the peak strength of NaCaHSiO_4 in the residue phase with NaOH concentration of 50% is much higher?

Response: Thank you very much to point out the faults in our manuscript. We are very sorry for our negligence of misuse an unfinished XRD diagram in Figure 4. We have corrected it in the revised manuscript.

Comment 4: In the analysis of the effect of alkali to gangue ratio on the extraction rate of aluminum, describes that with the alkali to gangue ratio rising, $\text{Na}_8\text{Al}_6\text{Si}_6\text{O}_{24}(\text{OH})_2(\text{H}_2\text{O})_2$ and $\text{Ca}_2\text{Al}_2\text{SiO}_6(\text{OH})_2$ peaks became weaken and peaks of NaCaHSiO_4 became stronger. But why is the peak strength of the $\text{Na}_8\text{Al}_6\text{Si}_6\text{O}_{24}(\text{OH})_2(\text{H}_2\text{O})_2$ and $\text{Ca}_2\text{Al}_2\text{SiO}_6(\text{OH})_2$ phases at a ratio of 6 is significantly higher than the peak at the ratio is of 5?

Response: We are very sorry that due to our negligence some of the samples submitted for XRD tests were labeled incorrectly, resulting in some errors in the figure 6. We have reanalyzed the samples and corrected them in the revised manuscript.

Comment 5: In XRD analysis, as the reaction temperature rising from 200 to 260 °C . The peaks of $\text{Na}_8\text{Al}_6\text{Si}_6\text{O}_{24}(\text{OH})_2(\text{H}_2\text{O})_2$ and $\text{Ca}_2\text{Al}_2\text{SiO}_6(\text{OH})_2$ became weaker, But why is the peak strength of the $\text{Na}_8\text{Al}_6\text{Si}_6\text{O}_{24}(\text{OH})_2(\text{H}_2\text{O})_2$ and $\text{Ca}_2\text{Al}_2\text{SiO}_6(\text{OH})_2$ phases at 220°C higher than the peak when the reaction temperature is at 200°C ?

Response: Thank you for your careful reading of our manuscript. We are sorry that the meaning of this paragraph is incomplete. For better understanding, we have added some sentences to this paragraph.

When the reaction temperature was at 200°C, the phases of the residue had transformed. Characteristic peaks of kaolinite existing in XRD patterns of ARCG disappeared, and new peaks of $\text{Na}_8\text{Al}_6\text{Si}_6\text{O}_{24}(\text{OH})_2(\text{H}_2\text{O})_2$, $\text{Ca}_2\text{Al}_2\text{SiO}_6(\text{OH})_2$, $\text{Ca}(\text{OH})_2$ and NaCaHSiO_4

emerged in the leached residue. When the reaction temperature reached 220 °C , the peaks intensity of $\text{Na}_8\text{Al}_6\text{Si}_6\text{O}_{24}(\text{OH})_2(\text{H}_2\text{O})_2$ and $\text{Ca}_2\text{Al}_2\text{SiO}_6(\text{OH})_2$ had an increasing tendency, which similar to the previous research in treating fly ash [31, 33]. When the temperature increased from 220 °C to 260 °C , the peaks of $\text{Na}_8\text{Al}_6\text{Si}_6\text{O}_{24}(\text{OH})_2(\text{H}_2\text{O})_2$ and $\text{Ca}_2\text{Al}_2\text{SiO}_6(\text{OH})_2$ became weaker and disappeared. Finally, the NaCaHSiO_4 became the only stable phase, corresponding to a better alumina extraction effect.

We have added the above discussion in the revised manuscript.

Comment 6: In Figure 7, No experiment was performed with a reaction temperature of 280°C or higher temperature. Why can it be stated that 260°C is the optimal reaction temperature?

Response: Thank you for your valuable comments. According to your helpful advice, we have added the data of 280°C into the revised manuscript.

Comment 7: In figure 11, the SEM image is not clear enough.

Response: Thank you for your instructive suggestions. We are very sorry that we are unable to get clearer images as our institute has suspended sample testing services since the pneumonia outbreak caused by the novel coronavirus.

Comment 8: Why Table 1 is represented by Al/Si and Table 2 is represented by A/S?

Response: Thank you for your valuable advice. According to the reviewer's comment, we have changed the "A/S" in Table 2 to "Al/Si" in the revised manuscript.

Comment 9: There are a lot of annotations, formatting, and language errors in the manuscript. For examples, in Figure 6、 Figure 8 and Figure 9,the author write NaCaHSiO_4 as NaCHSiO_4 .The author needs to carefully review and improve the manuscript.

Response: Thank you very much. According to your comment, we have had the manuscript polished and corrected the mistakes.

Reviewer 2:

Comment 1: This paper investigated the extraction of alumina from alumina rich coal gangue by a hydro-chemical process. However, it is believed that this article didn't give some valuable scientific discussion about the alumina extraction process. The organization and writing of article were extremely similar with the ref. [31], their previous research paper with similar hydro-chemical process to fly ash. Therefore, It's not suitable for the publication of the research article type.

Response: Thank you for your comments. This research is not a simple repetition of ref. [31] as alumina containing phases of ARCG and fly ash are different. The major alumina containing phases in ARCG is kaolinite different from that of fly ash, which indicates that the phase

transformations in the alumina extraction process from coal gangue and fly ash are different. Based on the ref. [31], we found that ARCG is easier to convert to NaCaHSiO_4 than fly ash, confirming that extraction of alumina from ARCG is more feasible. Moreover, the kinetic law of ARCG in alkaline hydrothermal systems has not been studied by previous researchers. According to the systematic experiments and analysis, this study show that the alumina leaching process is controlled by chemical reactions and the apparent activation energy is as high as 54.21 kJ/mol, which may explain why the temperature can greatly influence the efficiency of alumina. Therefore, we believe that this article is of great significance for the study of alumina extraction from coal gangue.

Reviewer 3:

Comment 1: The ARCG sample selected in this paper comes from a coal mine in Shanxi Province, but according to the chemical analysis and XRD results of the sample, the main component of the ARCG sample is kaolin, which seems to deviate from the theme of this paper. It is better to extract aluminum from kaolin than from ARCG.

Response: Thank you for your valuable comments. The gangue used in the experiment is a representative type of gangue in China, which is characterized by high content of kaolinite. The aim of this study is to promote the high added value utilization of this type of coal gangue.

Therefore, we hope that the topic may focus on the theme of coal gangue.

Comment 2: The purity and grade of reagents and water should be indicated in the experiment. P7 line 4.

Response: Thank you for your instructive suggestions. According to your helpful advice, we have rewritten this part in section 2.1, such as the following:

The reagents NaOH, Ca(OH)₂, Al(OH)₃ used in the experiment were of analytical purity grade offered by Sinopharm Chemical Reagents Co., Ltd. (Shanghai, China) and used without further purification. Ultrapure water was obtained from a Milli-Q water purification system (Millipore, USA).

Comment 3: In P7 line 36, does the content of Al₂O₃ refer to A₂O₃ in ARCG? If so, then the content of Al₂O₃ is fixed. According to the fixed value of Al₂O₃/NaOH, the amount of NaOH added in each experiment should be fixed.

Response: Thank you for your careful reading of our manuscript. In P7 line 36, Al₂O₃ is not Al₂O₃ in ARCG, but refers to Al₂O₃ in the liquid reaction medium (sodium aluminate solution). The molar ratio of Na₂O to Al₂O₃ in the reaction medium (sodium aluminate solution) is 25. Therefore, when the influence of the alkali to gangue ratio is investigated, the amount of NaOH added is changed, and when the influence of other factors is investigated, the amount of NaOH added is fixed.

Comment 4: In P6 line 58, the unit of granularity should be μm or mm.

Response: Thank you very much. According to your comment, we have corrected it in the new manuscript, such as the following:

Samples of ARCG were crushed to less than 2.36mm and dried in an oven at 105 °C for 24h.

Comment 5: In the experiment of Al₂O₃ extraction, the average value should be taken from the parallel experiment of multiple units, and the error range should be explained.

Response: Thank you for your instructive suggestions. According to your comment, we have added the relevant instructions in section 2.2 of the revised manuscript.

Comment 6: In Section 2.3, the experimental process of XRD is not clear, so the XRD analysis method in the reference “Comparative studies on catalytic mechanisms for natural chalcopyrite-induced Fenton oxidation: Effect of chalcopyrite type. Journal of Hazardous Materials 2020, 381, 120998” can be used.

Response: Thank you for your valuable comments. According to the reviewer’s suggestion, we have corrected it in the revised manuscript.